# Nanoparticles as a Tool in Neuro-Oncology Theranostics

**DOI:** 10.3390/pharmaceutics13070948

**Published:** 2021-06-24

**Authors:** Andrea L. Klein, Grant Nugent, John Cavendish, Werner J. Geldenhuys, Krishnan Sriram, Dale Porter, Ross Fladeland, Paul R. Lockman, Jonathan H. Sherman

**Affiliations:** 1The George Washington University School of Medicine and Health Sciences, Washington, DC 20037, USA; andreaklein@gwu.edu; 2West Virginia University School of Medicine, Morgantown, WV 26505, USA; gnugent1@mix.wvu.edu (G.N.); jzcav4@gmail.com (J.C.); 3Department of Neuroscience, West Virginia University School of Medicine, Morgantown, WV 26506, USA; werner.geldenhuys@hsc.wvu.edu (W.J.G.); prlockman@hsc.wvu.edu (P.R.L.); 4Department of Pharmaceutical Sciences, West Virginia University School of Pharmacy, Morgantown, WV 26506, USA; rf00015@mix.wvu.edu; 5National Institute for Occupational Safety and Health, Centers for Disease Control and Prevention, Morgantown, WV 26505, USA; kos4@cdc.gov (K.S.); dhp7@cdc.gov (D.P.); 6Department of Neurosurgery, West Virginia University School of Medicine, Morgantown, WV 26505, USA

**Keywords:** theranostic, imaging, neuro-oncology, neurosurgery, nanoparticles

## Abstract

The rapid growth of nanotechnology and the development of novel nanomaterials with unique physicochemical characteristics provides potential for the utility of nanomaterials in theranostics, including neuroimaging, for identifying neurodegenerative changes or central nervous system malignancy. Here we present a systematic and thorough review of the current evidence pertaining to the imaging characteristics of various nanomaterials, their associated toxicity profiles, and mechanisms for enhancing tropism in an effort to demonstrate the utility of nanoparticles as an imaging tool in neuro-oncology. Particular attention is given to carbon-based and metal oxide nanoparticles and their theranostic utility in MRI, CT, photoacoustic imaging, PET imaging, fluorescent and NIR fluorescent imaging, and SPECT imaging.

## 1. Introduction

The rapid growth of nanotechnology and the development of novel nanomaterials with unique physicochemical characteristics provides potential for the utility of nanomaterials in theranostics, including neuroimaging, for identifying neurodegenerative changes or central nervous system malignancy. Theranostic agents play an important and emerging role in the diagnostics and treatment of metastatic tumors, allowing for refinement and reduction of treatment intervention of the cancer patient, and the combination of theranostic agents with nanoparticles has been an area of active research in the past few years. The small size and large surface area of nanomaterials permits translocation across biological barriers and enhances the interaction with cellular and intracellular components of tumor cells and the tumor microenvironment. In any other context, such extravasation would be considered undesirable, especially when not targeted, but in the context of cancer theranostics this is a tremendously useful property. The current lack of literature investigating the modifications necessary to properly target these nanoparticles, especially to the neuro-oncology space, as well as the lack of literature on nanoparticles’ imaging visibility and interactions and the off-target toxic potential of such nanomaterials limits their effective clinical translation. For the CNS oncology scope, the brain poses several challenges for treatment, including the limitations on toxicity that could lead to neurodegeneration of native cells, thereby impacting patient mortality and morbidity greatly.

Theranostic agents for the CNS follow the same criteria to achieve clinically relevant levels in the brain or in a primary/metastatic tumor site. Figure 1 illustrates the general scope of nanoparticles that have been developed, each focused on specific characteristics that optimally allow for specific drug delivery. The use of nanoparticles allows for both the targeting of nanoparticles for use in imaging of tumor location and size, as well as delivery of a therapeutic agent, which could be a small chemotherapeutic agent or a radioactive isotope (Figure 1B,C). The traditional formulation system for nanoparticles in theranostic delivery is the packaging of a compound into a nanoparticle, alone or in conjunction with a targeting ligand, e.g., gadolinium and an antibody for targeting to a specific receptor (Figure 1D).

Here we present a systematic and thorough review of the current evidence pertaining to the imaging characteristics of various nanomaterials, their associated toxicity profiles, and mechanisms for enhancing tropism in an effort to demonstrate the utility of nanoparticles as an imaging tool in neuro-oncology. We explore in more detail similar types of reported work, with particular interest in the novelty of the present work, limitations, and possibilities.

## 2. Use of Nanoparticles as a Diagnostic Imaging Tool

This section describes the results of our literature search for studies regarding the use of nanoparticles as contrast agents with which a variety of medical imaging modalities can view a variety of tumor types. This should be prefaced with the understanding that nearly all of these studies were conducted to view tumors xenografted into a murine (mouse) model. Additionally, the vast majority of our results were descriptions of successful use, rather than head-to-head comparisons of nanoparticles. Those that did list comparative data were those looking at the effect of an addition to or modification of a base nanoparticle (i.e., A vs. A + B). Thus, it was not reasonable to draw conclusions about the utility of one nanoparticle vs. another.

### 2.1. MRI

Our review of the literature shows that magnetic resonance imaging (MRI) as an imaging modality has seen the largest variety of nanoparticles used as either carriers for agents that provide contrast or as direct contrast agents themselves. Single wall carbon nanotubes (SWCNT) are the first example of the former. These agents have been transfected with molecules known to provide contrast on MRI, such as manganese [4], gadolinium [5,6,7], and iron oxide [8,9,10], in order to visualize tumor types including 4T1 breast carcinoma [4,8,9,10] and S180 sarcoma [5,6,7]. In a similar molecular sense, multiwalled carbon nanotubes (MWCNT) have also been successfully utilized as carriers for iron oxide [11,12,13,14,15] and gadolinium [16]. These agents have aided in the visualization of xenografts consisting of KB cell tumors, human breast carcinomas (MCF-7 cell line), and hepatocellular carcinomas. Another carbon-based agent, the magnetic hollow porous carbon nanoparticle (MHPCN), is an interesting compound of carbon nanodots and iron oxide that has been studied as an MRI contrast agent for visualizing human cervical carcinoma [17]. Iron oxide has also been paired with graphene oxide nanosheets and has been successfully used to produce contrast in 4T1 breast carcinoma [18,19]. Whether these carbon-based particles and their paired metals act in synergism to enhance contrast is an interesting question. A study by Fu et al. demonstrated enhanced contrast on MRI when graphene oxide was transfected with iron oxide and compared with the contrast produced by graphene oxide alone [20]. Furthermore, Rammohan et al. showed that when gadolinium was paired with nanodiamonds, relaxivity was increased 10-fold compared with that of gadolinium alone [21].

Looking at agents that stray further from a carbon-based carrier, some of the aforementioned attached particles were popular in our search. Iron oxide has been used alone as an MRI contrast agent for imaging models of nasopharyngeal and breast carcinoma [22]. More frequently, iron oxide has served to provide MRI contrast while combined with other particles of varying utility. Examples include combination with iron-platinum nanoparticles in visualizing epidermoid and breast carcinoma [23], doping with zinc particles to improve contrast in breast carcinoma [24], formation of zinc–cobalt–ferrite nanoparticles to enhance the contrast effect seen with iron oxide alone in melanoma (C540 cell line) [25], and pairing with upconversion nanoparticles and gold particles in a dual modal imaging and photothermal therapy agent for breast carcinoma [26]. Perhaps the most common utilization of iron oxide as an MRI contrast agent has been as part of larger multi-agent nanoparticles with multimodal imaging and even therapeutic capability. To spare redundancy, the individual utility of the contrast agents in each example are listed in their respective imaging utility subsections later in this work, described as part of multimodal imaging nanoparticles also. Important to recognize are the many different particles iron oxide has been compounded with while still maintaining its ability to successfully provide contrast in tumors visualized with MRI. Iron oxide has filled this role in a number of combinations to include: silver sulfide and a near-infrared (NIR) fluorophore to visualize breast carcinoma [27]; tantalum oxide, copper sulfide, and zinc phthalocyanine to image cervical carcinoma (U14 cell line) [28]; fluorescent semiconducting polymer to image cervical carcinoma (HeLa cell line) [29]; dimercaptosuccinic acid, bevacizumab, and technetium-99m to view breast carcinoma [30]; bovine lactoferrin and VivoTag 680 to visualize colon carcinoma (Caco-2 cell line) [31]; upconversion nanoparticles (Y, Yb, Er) and squaraine dye to view 4T1 breast carcinoma [32]; polydopamine particles and DNA probes to image breast carcinoma (MCF-7 cell line) [33]; molybdenum disulfide nanosheets to view breast carcinoma [34]; and polypyrrole to again view breast carcinoma [35].

Gadolinium as we know is commonly used in medical imaging as an MRI contrast agent. It has thus been employed for this use in a number of nanoparticle compounds other than the carbon-based carriers. In a study using titanium dioxide for sonodynamic therapy of prostate adenocarcinoma (LNCaP cell line), gadolinium was attached specifically to provide MRI utility [36]. It served the same purpose when attached to a zinc oxide quantum dot template that was being used for fluorescent imaging of a pancreatic carcinoma (BxPC-3 cell line) [37]. Using a keratin template, gadolinium has been combined with manganese dioxide to achieve an enhanced T1-weighted MRI effect in 4T1 breast carcinoma due to optimal structure on the template and the combination effect of two MRI contrast agents [38]. Zhong et al. created a scintillating nanoparticle composed of NaCeF_4_:GdTb and proposed that the presence of lanthanide Ce and Tb ions actually enhanced the MRI contrast capability of gadolinium in lung carcinoma (A549 cell line) [39]. Similarly, Wang et al. added gadolinium to expand the multimodal imaging capabilities of their upconversion nanoparticle NaYF_4_/NaGdF_4_ to enhance MRI visualization of cervical carcinoma (HeLa cell line) [40].

Manganese is an element that displays utility as an MRI contrast agent, often as the Mn^2+^ ion or as manganese dioxide. It has actually been reported that manganese dioxide will react with increased H^+^ and GSH found in the tumor microenvironment to yield Mn^2+^, increasing the observed T1-weighted relaxivity [38,41]. As such, a list of studies exists in which either is utilized to add MRI functionality to the designed nanoparticle. Mn^2+^ has been added to a calcium carbonate carrier along with chlorin e6 and a polydopamine coat to view 4T1 breast carcinoma [42]; added to a polydopamine carrier along with indocyanine green and doxorubicin for theranostic study of 4T1 breast carcinoma [43]; combined with polydopamine coated black titanium dioxide and chlorin e6 for theranostic study of 4T1 breast carcinoma [44]; combined with gold, titanium dioxide, and doxorubicin for theranostic study of cervical carcinoma (HeLa cell line) [45]; fabricated with a near-infrared dye to form a nanoscale metal-organic particle and coated with polydopamine for theranostic study of breast carcinoma [46]; combined with collagenase with the goal of degrading the tumor ECM for better perfusion and imaging of breast carcinoma [47]; and combined with calcium carbonate and doxorubicin for theranostic study of breast carcinoma [48]. Manganese dioxide has been combined with chlorin e6 and doxorubicin for theranostic study of 4T1 breast carcinoma [41]; and attached to keratin carrier surface along with gadolinium oxide and doxorubicin for theranostic study of 4T1 breast carcinoma [38]. Two unique applications of manganese were also uncovered in our search. One of which used manganese in combination with tungsten to create a bimetallic oxide (MnWOx) for the purpose of sonodynamic therapy enhancement as well as imaging of 4T1 breast carcinoma [49]. The other used manganese sulfide (MnS) and combined it with zinc sulfide and gold for theranostic study of breast carcinoma [50].

Finally, there are two miscellaneous molecules that, while less popular, have been incorporated into nanoparticles for purposes of providing contrast on magnetic resonance imaging. Titanium dioxide has been combined with Yb, Ho, and F to form an upconversion nanoparticle useful for imaging breast carcinoma (MCF-7 cell line) [51], and vanadium disulfide nanodots have been paired with technetium-99 for theranostic study of 4T1 breast carcinoma [52].

Briefly, we verify that various nanoparticles have been successfully used to produce contrast in brain tumor models as well. One review describing the imaging of various glioma/glioblastoma models reported the use of several agents to produce contrast on MRI, including gadolinium, iron oxide, and gold nanoparticles [53]. In another review, various brain tumors are reported to have been imaged using iron oxide, gadolinium, and manganese oxide nanoparticles [54].

### 2.2. CT

Computed tomography (CT) as an imaging modality did not yield as many results in our search as those for MRI, perhaps because of the sharp image quality MRI provides, particularly for soft tissues (i.e., tumors). Nonetheless, a number of studies were found using a variety of miscellaneous nanoparticles as contrast agents under CT imaging of tumors. Some of these were used specifically for CT imaging, but many were constructed for multimodal imaging or theranostic use. Several of the lanthanide elements have been incorporated into two examples of nanoparticles used for CT imaging. The same scintillating nanoparticle composed of Ce, Gd, and Tb ions found useful for MRI also provided contrast for CT imaging of lung carcinoma (A549 cell line) [39]. Similarly, the aforementioned Gd containing upconversion nanoparticle NaYF_4_/NaGdF_4_ used for MRI also provided CT contrast for cervical cancer (HeLa cell line) [40]. Titanium compounds have been rendered useful for CT contrast as well. Growing gold on the surface of titanium carbide allowed CT imaging of 4T1 breast carcinoma [55], and doping titanium dioxide with tungsten allowed CT imaging of the same [56]. The MnWOx nanoparticle that had MRI capability also saw utility as a CT contrast agent from the presence of tungsten [49]. Iron oxide was useful in two multimodal imaging studies that included CT imaging. One combined iron oxide with bovine lactoferrin, alginate-enclosed chitosan-encapsulated calcium phosphate, and Vivotag 680 for theranostic study of colon cancer (Caco-2 cell line) [31]. The other included an NIR-fluorophore and silver sulfide that served to enhance the CT contrast effect of iron oxide in breast carcinoma [27]. Tantalum oxide doped with iron, presumably with combined effect, provided CT contrast ability to the multimodal nanoparticle also consisting of copper sulfide and zinc phthalocyanine for theranostic study of cervical carcinoma (U13 cell line) [28]. Rhenium sulfide was used as a lone agent to view 4T1 breast carcinoma [57]. A bismuth–carbon nanoparticle composite was constructed as a naïve compound for imaging cervical carcinoma (HeLa cell line) [58]. Nanoscale coordination polymers of hafnium and bis-alkene as CT contrast agents were compounded with doxorubicin and chlorin e6 for theranostic study of breast carcinoma [59]. Finally, our search yielded a study using ExiTron nano, an alkaline earth-based nanoparticulate contrast agent manufactured by Viscover, to view liver metastases of an unspecified tumor of origin [60]. In terms of brain-tumor-specific imaging, gold nanoparticles have been successfully used as CT contrast agents in a U87 malignant glioma model [53].

### 2.3. Fluorescent and NIR Fluorescent Imaging

Here we describe findings for fluorescent and NIR fluorescent imaging, the difference being the use of light in the near-infrared (NIR) spectrum vs. shorter wavelengths. NIR fluorescence carries the advantages of deeper tissue penetration and less autofluorescence from surrounding tissues (low background), making it the intuitively preferred modality for tumor imaging [61]. Nonetheless, our search yielded studies describing the use of each.

Zinc oxide has been useful as a fluorescent imaging agent in two forms from our search. Zinc oxide quantum dots added fluorescent imaging utility to the multimodal nanoparticle used for theranostic study of pancreatic cancer (BxPC-3 cell line) [37], and a hollow zinc oxide nanocarrier of paclitaxel and folate allowed fluorescent imaging in a theranostic study of breast carcinoma [62]. Iron oxide nanoparticles are not used directly as agents for fluorescent imaging but rather can serve as the core particle and thus the carrier for agents that provide fluorescence. In one multimodal imaging study, iron oxide was encapsulated with a semiconducting polymer that provided fluorescent capability for viewing cervical cancer (HeLa cell line) [29]. A multimodal combination of iron oxide and an upconversion nanoparticle served as a carrier for squaraine dye, an agent used for downconversion fluorescent imaging of 4T1 breast carcinoma [32]. Taking advantage of the fluorescent capacity of zinc, an iron oxide–polydopamine nanoparticle was transfected with a DNA probe tasked with increasing intracellular release of endogenous zinc for imaging of breast carcinoma (MCF-7 cell line) [33]. Manganese dioxide has served a similar role in a multimodal theranostic study using a hollow manganese dioxide nanoplatform as a carrier for doxorubicin and chlorin e6 as the fluorescent agent to view 4T1 breast carcinoma [41]. The multimodal MnWOx nanoparticle used for theranostic study of 4T1 breast carcinoma was also turned into a fluorescent agent due to the attachment of DiR iodide dye [49]. Graphene oxide has served as carrier for attachments of iron oxide and the fluorescent agent cyanine 5 for viewing 4T1 breast carcinoma [19]. Titanium dioxide doped with Yb, Ho, and F showed utility for upconversion fluorescent imaging of breast carcinoma (MCF-7 cell line) [51]. A titanium dioxide polypyrrole nanoparticle conjugated with DiR fluorescent dye was used to successfully image ovarian carcinoma (Skov3 cell line) [63]. The previously discussed scintillating nanoparticle composed of NaCeF_4_:GdTb was endowed with X-ray excited fluorescence for imaging of lung carcinoma (A549 cell line) due to the presence of terbium ions [39]. SWCNT when used for fluorescent imaging have been conjugated with a fluorophore such as chlorin e6 when viewing squamous cell carcinoma [64] or a cyanine 5-labeled aptamer to view xenograft tumors of ALL or Burkitt’s lymphoma [65]. Finally, small molecule gold nanoparticles found use in a study viewing lung carcinoma [66].

SWCNT have been studied extensively as NIR fluorescent imaging contrast agents, occasionally with a fluorescent dye attached or a structural change. This is due to their good optical absorbance in the NIR region [67], though only a subset of nanotube chiralities will actually fluoresce or heat well under a NIR laser on their own [68]. A few examples of naïve SWCNT, or those without dye or structural alterations, include isolates of these chiralities. These are typically used in the NIR-II region. Diao et al. viewed the vasculature of 4T1 breast carcinoma by specifically isolating the (12, 1) and (13, 3) chiralities of SWCNT which demonstrate ~5-fold higher photoluminescence than unsorted SWCNT [69]. Antaris et al. isolated the (6, 5) chirality of SWCNT to view 4T1 breast carcinoma, and this isolate displayed 6-fold brighter luminescence than unsorted SWCNT [68]. Two studies chose to stabilize unsorted SWCNT for NIR-II imaging with an M13 bacteriophage, and found that this allowed the unsorted nanotubes to provide good fluorescence for viewing prostate adenocarcinoma [70] and ovarian carcinoma (OVCAR8 cell line) with tumor-to-background ratio actually 24- and 28-fold higher than standard NIR fluorescent dyes AF750 and FITC [71]. Another study took advantage of both a nanofluorophore and SWCNT without actually combining them. In this work, unsorted SWCNT were used to image a 4T1 breast carcinoma tumor via the enhanced permeability and retention effect, and a nanofluorophore called p-FE was created to visualize the tumor vasculature. These agents emitted two different colors of fluorescence, and the unsorted SWCNT had to be given at a high dose with 5-fold longer imaging exposure time and 2-fold larger pinhole on the imaging device to capture fluorescence emission [72]. The intrinsic NIR properties of unsorted SWCNT were used alone with no fluorescent dye in the imaging of breast carcinoma [73]. Functionalization via structural change to both SWCNT and graphene nanosheets with the addition of poloxamer 407 has been used to image squamous cell carcinoma by NIR fluorescence [74]. A semiconducting SWCNT with large diameter has been used to view breast carcinoma as well as cerebrovascular flow [75]. Several studies have employed the attachment of a fluorescent dye to enhance NIR fluorescent capability of SWCNT. These include the addition of Alexa Fluor 594 to view ovarian carcinoma [76], cyanine 7 to view pancreatic carcinoma (BxPC-3 cell line) [67], cyanine 5.5 to view breast carcinoma (MCF-7 cell line) [77], IR-783 to view sarcoma (S180 cell line) [6], and DiR to view sarcoma (S180 cell line) [7].

Several other agents have been employed for NIR fluorescence study as well, many being fluorescent dyes expanding the imaging capability of other nanoparticles. Graphene oxide with an attachment of cyanine 5.5 has been used for imaging 4T1 breast carcinoma [78]. Multimodal iron oxide has been rendered useful for NIR fluorescent imaging via attachment of the fluorescent dye VivoTag 680 to image colon carcinoma (Caco-2 cell line) [31]. The NIR fluorophore DiR has been incorporated into a PEGylated phospholipid mixed micel also containing iron oxide and silver sulfide in order to expand imaging capabilities of breast carcinoma [27]. The NIR fluorescent dye IR825 has been attached to manganese nanoscale metal–organic particles [46] and to human serum albumin [79] for imaging of breast carcinoma in both cases. Chlorin e6 was attached to a multimodal calcium carbonate carrier for imaging of 4T1 breast carcinoma [42]. Titanium dioxide was rendered capable of NIR fluorescent imaging of tumors, interestingly by doping with Nb^5+^ ions, which caused the molecule’s light absorption capacity to shift into the NIR region [80]. One study actually simplified the approach, citing problems of low brightness and low fluorescence quantum yield of previous carrier-based systems, with the NIR fluorescent dye NPAPF, which was prepared for administration alone with no carrier or attachments to image breast carcinoma [81]. Another study used a lone downconversion nanoparticle (NaYYbErF), attachments being molecules for tumor targeting and biostability, to view ovarian carcinoma (COV362 cell line) [82]. Finally, one study employed the use of fluorescent CdTe quantum dots in order to image KB tumor [83].

Various agents have been used specifically for fluorescent imaging of brain tumors as well. Glioma/glioblastoma models have been imaged using quantum dots [53,84], SWCNT [85], liposomal nanoparticles, and holotransferrin nanoparticles [53]. Polyacrylamide, iron oxide with Cy5.5 dye, and quantum dots again were reported as agents used to image various tumors and tumor vasculature. It should be noted, however, that the skull proves to be a significant physical barrier to fluorescent imaging, and the utility of this modality is mainly isolated to intraoperative localization of brain tumor tissue [54].

### 2.4. Photoacoustic Imaging

SWCNT are popular contrast agents for photoacoustic (PA) imaging as well, again secondary to their responsiveness to light in the NIR region [64,86]. Although some studies attached agents to SWCNT in order to enhance PA imaging, SWCNT were the sole mode of contrast in a study imaging squamous cell carcinoma [64]. One study took the approach of attaching PA contrast dyes to SWCNT, creating five separate “flavors” of nanoparticles, those including QSY_21_ and indocyanine green exhibiting over 100-fold higher PA contrast than SWCNT alone [87]. Another study shared the mechanism of indocyanine green attachment for the imaging of 4T1 breast carcinoma, notably showing 2-fold greater enhancement with the SWCNT–indocyanine green combination than indocyanine green alone [88]. Other carbon-based nanoparticles useful for PA imaging include graphene oxide nanosheets, which served as PA contrast in two multimodal studies imaging 4T1 breast carcinoma [19,89], and hollow mesoporous carbon nanospheres, which did the same for two other theranostic xenograft studies [90,91]. Aside from carbon-based nanoparticles, our search yielded a mix of other agents that have been studied as PA contrast. Titanium dioxide has been modified several times to shift its absorption into the NIR range and to thus become useful for PA imaging. One example is from a previously mentioned study that utilized doping of titanium dioxide with tungsten in order to visualize 4T1 breast carcinoma [56]. Making titanium dioxide an oxygen-deficient molecule (TiO_2_-x) also increased absorption in the NIR range, allowing PA imaging [92]. The same study that doped titanium dioxide with Nb^5+^ ions for NIR fluorescent imaging found this to be useful for creating a PA contrast agent as well [80]. Gold nanorods display good absorption in the NIR range and thus have been used as PA contrast to image cervical carcinoma (HeLa cell line) [93]. Seeding of gold actually allowed functionalization of a titanium carbide nanosheet carrier for the imaging of 4T1 breast carcinoma [55]. The previously mentioned multimodal imaging study with calcium carbonate also reported good PA imaging of 4T1 breast carcinoma due to calcium carbonate having a polydopamine coat that expanded its utility [42]. MoS_2_–iron oxide, a nanocomposite that showed utility as an MRI contrast agent, was also found to be useful for PA imaging of breast carcinoma [34]. In a theranostic study of cervical carcinoma (U14 cell line), a bismuth sulfide–manganese oxide nanocomposite served as the contrast agent [94]. The MRI contrast vanadium disulfide nanodots discussed earlier also show strong NIR absorbance and provide PA contrast of 4T1 breast carcinoma as well [52]. Finally, an interesting coordination polymer nanodot composed of ruthenium ions and phenanthroline, neither of which show significant optical absorbance alone, showed strong NIR absorbance as a compound allowing PA imaging of 4T1 breast carcinoma [95].

For brain tumor imaging, holotransferrin nanoparticles have served as PA contrast agents to view glioma models [53], and gold nanoparticles have been useful for viewing a variety of brain tumors [54]. Several studies have also utilized SWCNT for PA imaging of glioma/glioblastoma models [86,96,97]. Because of their close mechanistic relationship, PA and fluorescent imaging share the limitation imposed by the physical barrier of the skull.

### 2.5. PET Imaging

Due to the nature of the imaging mechanism, studies that included PET imaging of tumors attached a radiolabel to the nanoparticle under investigation. All but one chose the radioisotope [64] Cu. The study that differed used both 1,4,7,10-tetraazacyclododecane-1,4,7,10-tetraacetic acid (DOTA) and desferrioxamine B (DFO) as radiolabels attached to SWCNT for imaging colon adenocarcinoma (LS174T cell line) [98]. All of the following nanoparticles were transfected with [64] Cu for PET imaging: graphene oxide nanosheets in two studies viewing breast carcinoma [89,99], zinc oxide to image breast carcinoma [100], boron nitride nanoparticles in a therapeutic study of breast carcinoma [101], and molybdenum disulfide–iron oxide nanocomposite as previously mentioned to view breast carcinoma [34]. Iron oxide radiolabeled with [64] Cu has been shown to provide contrast on PET for imaging various brain tumors [54], while SWCNT have been used to view a glioblastoma model [96].

### 2.6. SPECT Imaging

As with PET imaging, the use of the SPECT modality necessitates the addition of a radioisotope to the nanoparticle under study. Technetium-99 was the isotope of choice in most studies including the use of iron oxide [30], vanadium disulfide [52], and gallic acid–ferric nanocomplexes [102] to view breast carcinoma. Ref. [103] Iodine was the radioisotope attached in a multimodal imaging study using SWCNT to view breast carcinoma [4].

### 2.7. Miscellaneous

The remaining studies our search yielded cover a range of imaging modalities that were not as popular in the literature as those already mentioned. Only one study utilized X-ray imaging and did so by using gold as an X-ray absorber attached to graphene oxide for imaging breast carcinoma [18]. Two studies looked at ultrasound imaging, one of which used MWCNT as a contrast agent to visualize prostate carcinoma (CP-3 cell line) [104]. The other utilized pulsed magneto-motive ultrasound with zinc-doped iron oxide to provide magnetization and contrast for an epidermoid carcinoma (A431 cell line) [105]. A multimodal study utilized MWCNT as a contrast agent for MRI as well as microwave-induced thermoacoustic imaging of breast carcinoma [15]. Ref. [103] Iodine was attached to reduced graphene oxide nanoparticles for gamma imaging and IR thermal imaging of breast carcinoma [106]. Other nanoparticles used for IR thermal imaging include SWCNT [4] and iron oxide-polypyrrole [35], which both allowed visualization of breast carcinoma. A multimodal nanoparticle consisting of iron oxide, silver sulfide, and a NIR fluorophore successfully provided contrast for mammography due to the presence of silver sulfide [27]. Finally, magnetic particle imaging (MPI) was a modality explored with multimodal janus iron oxide (Fe_3_O_4_) to image cervical carcinoma. It was stated that the crystallinity of janus iron oxide allowed it to provide increased contrast on MPI compared with plain iron oxide (Fe_2_O_3_) [29].

## 3. Toxicity of Nanoparticles When Administered Systemically

Nanomaterials have been shown to cause various forms of systemic toxicity in animal and human studies. The systems affected and level of toxicity vary widely depending on nanoparticle physicochemical properties, size, shape, and route of exposure. They exert toxicity on multiple organs, including the lungs, heart, vasculature, blood, liver, and brain. The mechanisms of their toxicity are not fully understood but may be related to physical disruption of cells, induction of a pro-inflammatory state, and generation of reactive oxygen species (ROS).

### 3.1. Carbon-Based Nanoparticles

Carbon nanoparticles include single-walled carbon nanotubes (SWCNT), multi-walled carbon nanotubes (MWCNT), and ultrafine carbon black (UFCB). They have the potential to cause toxicity due to their small size, biopersistence, and fibrous nature, which can cause an asbestos-like reaction [107]. Toxicity following exposure via inhalation has been extensively documented, but further routes of exposure, including transdermal, ingestion, and intravenous administration, are less understood. Mouse studies showed that after inhalation exposure, MWCNT distributed in the lungs and later spread to other sites, including the tracheobronchial lymph nodes, diaphragm, brain, liver, and kidneys, presumably via hematogenous spread [108]. Additionally, amyloid fibril deposits were seen in spatial association with MWCNT aggregates and macrophages in several organs, including the brains, of CD1 mice following environmental inhalation [109].

Local toxic effects in target organs following carbon nanoparticle exposure have been documented. Deposition of SWCNT and MWCNT in the lungs can cause local inflammation, ROS generation, granuloma formation, interstitial fibrosis, and pneumonia [110,111,112,113]. Inhalation of UFCB also caused pro-inflammatory changes in the respiratory system; rats exposed to UFCB showed increased neutrophils and macrophage inflammatory protein-2 mRNA in bronchealveolar lavage samples [114]. Intranasal exposure to UFCB in mice induced airway inflammation and increased allergic sensitization to ovalbumin [115]. Within the blood, SWCNT and UFCB may induce platelet aggregation and prothrombotic effects [116,117]. Endovascular SWCNT and MWCNT may also bind to blood proteins, activate the complement system, activate immune cells, and cause hemolysis of red blood cells [118]. MWCNT caused accelerated plaque development in the aorta of apolipoprotein E(−/−) mice after repeated endotracheal exposure, which was possibly mediated through increased monocyte adhesion to endothelial cells and increased oxidative stress in monocytes [119]. Carbon nanoparticles have also demonstrated cardiotoxic effects. MWCNT of various forms exacerbated myocardial ischemia/reperfusion injury after oropharyngeal aspiration in mice [120]. The cardiotoxic effects are possibly due to induction of inflammation and ROS generation, as SWCNT increased cardiac lactate dehydrogenase and myeloperoxidase while reducing thiols in mice after pharyngeal aspiration [121]. Mice exposed to UFCB via endotracheal instillation showed reduced heart rate variability, a marker of autonomic nervous system function [122]. Intraperitoneally injected MWCNT functionalized with single-stranded DNA induced pro-inflammatory changes in the liver and plasma [123]. SWCNT accumulated in the liver of mice and caused elevations in aspartate aminotransferase, alanine aminotransferase, and bilirubin [124]. Carbon nanoparticles also exhibit genotoxic effects. Intraperitoneal administration of MWCNT increased chromosomal abnormalities in bone marrow leukocytes of mice [125]. More recent studies have shown neurological, pulmonary, and systemic effects following long-term inhalation of MWCNT. Increased mitochondrial ROS production, lipid peroxidation, mitochondrial swelling, and enhanced cytochrome *c* release has been reported in various brain regions following a two-week repeated inhalation exposure to MWCNT [126]. Increased respiratory and systemic effects were observed following a month-long exposure to MWCNT [127]. Prolonged inhalation of MWCNT has also been shown to modulate global gene and protein expression in the lungs [128].

### 3.2. Metal Oxide Nanoparticles

Metal oxide nanoparticles, including CuO, ZnO, TiO_2_, SiO_2_, and FeO have been utilized in various industrial and biomedical applications but have exhibited toxicity in vitro and in vivo.

Following intravenous injection of TiO_2_ nanoparticles in rats, significant accumulation of nanoparticles was found in the liver, spleen, lung, and kidney but less so the brain, lymph nodes, blood cells, or plasma [129]. Another study showed deposition of TiO_2_ nanoparticles in the lung, spleen, and liver of rats following intratracheal instillation but no deposition in brain, kidneys, or blood [103]. Intratracheal instillation of TiO_2_ nanoparticles in rats caused a reduction in body weight with disproportionate reduction in heart weight, liver edema, and elevations in blood urea nitrogen and aspartate aminotransferase [130]. Liver accumulation of TiO_2_ nanoparticles was seen after intraperitoneal injection in mice, with associated inflammatory response, hepatocyte apoptosis, and liver dysfunction [131]. Intratracheal instillation of TiO_2_ nanoparticles caused systemic immune activation in rats, with spleen congestion and increases in B lymphocytes [132]. Apolipoprotein E(−/−) mice exhibited increased atherosclerosis six weeks after intratracheal instillation of TiO_2_ nanoparticles, possibly related to pro-inflammatory changes and alterations in cholesterol metabolism [133]. Inhalation of TiO_2_ nanoparticles increased blood–brain barrier (BBB) permeability in aged rats and was associated with an increase in inflammatory cytokines within the brain [134]. Exposure of SH-5YSY neuroblastoma cells in vitro to TiO_2_ nanoparticles caused induction of ROS, autophagy, and apoptosis [135]. Intranasal delivery of TiO_2_ nanoparticles resulted in systemic uptake and caused allergic airway inflammation in a mouse model of asthma [136].

ZnO nanoparticles distribute to various organs, depending on the route of administration. Orally administered ZnO nanoparticles were found in the liver, spleen, and kidneys of mice, whereas they were found in the liver, spleen, kidneys, lungs, testes, and heart following intraperitoneal injection [137]. ZnO fume inhalation is known to cause metal fume fever in humans, but ZnO nanoparticles may also be toxic to various organs through different routes of exposure. ZnO nanoparticles irreversibly damaged primary cultured rat alveolar endothelial cell monolayers in vitro, decreasing mitochondrial function, increasing intracellular ROS, and increasing lactate dehydrogenase release [138]. Damaged alveolar endothelial cell monolayers lost their structural integrity and allowed nanoparticles to translocate from the apical to the basolateral fluid. Intratracheally instilled ZnO nanoparticles induced acute phase response and increased neutrophils in BAL fluid in mice [139]. Healthy human subjects showed a dose-dependent increase in c-reactive protein and serum amyloid A 24 h after inhalation exposure to ZnO nanoparticles [140]. Exposure of human coronary artery endothelial cells to ZnO nanoparticles in vitro resulted in reduced viability and increase in inflammatory markers such as IL-6 [141]. In another study, ZnO nanoparticles were added to a co-culture of human coronary artery endothelial cells and an alveolar epithelial cell line. IL-8, TNF-alpha, heme oxygenase-1, and platelet endothelial cell adhesion molecule-1 production was significantly increased, but this effect was blocked by co-administration with cytochalasin B, a phagocytosis inhibitor [142]. In addition to pulmonary and cardiovascular toxicity, ZnO nanoparticles may also be immunotoxic. Mice fed ZnO nanoparticles had reduced natural killer cell activity and NO production from splenocyte culture supernatant [143]. Orally administered ZnO nanoparticles caused anemia in rats, in addition to adverse effects on the stomach, pancreas, and retina [144]. Sub-chronic inhalation of ZnO nanoparticles caused pulmonary cell response changes in mice and altered the expression of ZN homeostasis-related genes [145]. Repeated intratracheal instillation of ZnO nanoparticles to monkeys has been shown to cause lung damage and systemic inflammation [146].

The toxicity of FeO nanoparticles is still under debate but overall appears to be lower than nanoparticles of other types. After being absorbed, FeO nanoparticles distribute systemically, mainly to the liver, but also to the brain, lungs, spleen, and bone marrow [147]. Fe_2_O_3_ nanoparticles reduced viability by over 50% when applied to mouse and human macrophage and alveolar epithelial cell lines in vitro [148]. In another study, application of FeO nanoparticles to a human alveolar type II-like cell line caused only minimal toxic effects with regard to mitochondrial dysfunction, DNA damage, and cell viability [149]. FeO nanoparticles caused no increase in inflammatory response when applied to human aortic endothelial cells [150].

## 4. Maximizing Nanoparticle Targeting to the CNS Tumor Microenvironment

Tumor tropism is defined as the propensity of an object or material to move towards the site of a tumor. For the purposes of this discussion, tumor tropism as it pertains to nanoparticles targeting CNS tumors will serve as the focus of discussion.

When administered systemically, unmodified nanoparticles do not inherently localize to tumors. It is the prerogative of those designing the nanoparticle to understand the tumor microenvironment and the modifiable properties of nanoparticles that can be used to improve tumor tropism by targeting the tumor microenvironment. By improving the targeting ability of the nanoparticle of investigation, an appropriately designed nanoparticle with sufficient imaging characteristics may therefore be used as a theranostic tool. The nanoparticle surface chemistry modifications necessary to make this possible are the subject of this section.

### 4.1. CNS as a Tumor Microenvironment

As with other tumors, the CNS tumor microenvironment can be divided into two components: structural and cellular. Structural components of the tumor microenvironment include the extracellular matrix and the blood–brain barrier, while cellular components include the tumor-associated macrophages, lymphocytes, fibroblasts, and stromal cells. The structural and cellular components work together to create a difficult-to-access and largely immunoprotected environment surrounding the tumor [151].

Cancer-associated fibroblasts generate dense, collagen-rich ECM that provides a mechanical and chemical barrier for nanoparticle delivery. Stromal cells actively promote tumor growth and metastatic dissemination by secreting signaling molecules, producing and remodeling the ECM, and coordinating cancer angiogenesis. Ref. [152] The blood–brain barrier serves as a unique chemical and physical filtration barrier specific to CNS malignancy and is perhaps one of the most significant challenges to overcome in early tumor detection, when BBB dysfunction may not yet be substantial enough to aid in nanoparticle delivery [153]. The dynamic and faulty vasculature of solid CNS tumors provides a “one-way” pressure valve that forces nanoparticles into the interstitium and traps them; solid tumors have a high interstitial fluid pressure that spontaneously drops to regular values to generate this gradient [154,155].

A difference between CNS tumors and other cancer types is that the brain is protected by a specialized vascular system, the blood–brain barrier (BBB). The BBB is a microvascular unit comprising vascular endothelial cells, astrocytes, pericytes, and neuronal attachments (Figure 2). Due to the occurrence of tight junctions between the vascular endothelium cells, a selectively permeable barrier is formed that prevents the passive permeability of compounds to the brain. In contrast, the peripheral vasculature is relatively leaky with the openings or “fenestrae” allowing for ready passive diffusion and compound distribution to the tissues. Figure 1 shows a diagrammatic representation of the BBB and peripheral vasculature. Of note, the kidney also contains this tight junctional system. This lack of BBB permeability by compounds is suggested to contribute to the high rate of clinical failure of potential therapeutic compounds for the CNS. The prediction of BBB permeability has been a focus of intensity study, with use of computational modeling to develop predictive models for BBB permeability. In general, compounds with a logBB > 0.3 are classified as crossing readily, while logBB < –1 are largely impermeable. Here the logBB is the ratio of [brain]/[plasma] of a compound. The use of nanoparticles to deliver a therapeutic modality to the brain has been shown to augment the pharmacokinetic behavior of compounds by increasing the distribution to the brain. For instance, a targeting peptide can be conjugated to the outside surface of the nanoparticle, which can use transporters located on the vascular endothelium to deliver the nanoparticle to the brain. This Trojan Horse system has been successfully used in several studies (Table 1).

M2 polarized macrophages and regulatory T lymphocytes are consistently found in the tumor microenvironment and are well-established anti-inflammatory mediators. There is also an increased preponderance of CD4+ T cells as opposed to CD8+ T cells in the tumor microenvironment, as well as a reduction in HLA-DR expression. These combined anti-inflammatory mediators with the associated additional increase of TGF-β secretion render the tumor microenvironment relatively protected from systemic immune surveillance [163]. Of note, recent evidence suggests that nanoparticles may use T cell-mediated transportation as a means of crossing the blood–brain barrier [164].

### 4.2. Using the Tumor Microenvironment to Improve Tumor Targeting for Nanoparticle Delivery

While the tumor microenvironment offers a unique challenge to nanoparticle delivery, one of the properties that makes nanoparticles such an attractive modality is the ability of the user to modify their surface chemistry and employ stimuli-response systems that take advantage of the properties of the tumor microenvironment to improve their tumor tropism. It should be noted that modifications to nanoparticles must serve a dual purpose: to increase the likelihood of the nanoparticle arriving at the tumor site and to not be discovered by the patient’s immune system before arriving. A careful ratio calculation and many attempts at trial and error will be required to find the balance between a surface chemistry that is similar enough to the tumor to encourage tropism while not being too foreign and generating an immune response that prohibits the nanoparticle from arriving at its destination. Nanoparticle formulation has evolved over the past few years, although many of the basic principles are still involved that were initially described by groups such as that of Robert Langer, Ph.D. Several different systems have been developed, including liposomal vesicles, emulsion-type, or solid-based nanoparticles. In general, the formulation consists of a polymeric system, which entraps or captures a cargo of interest.

There are two main methods of designing surface chemistry that targets a specific tumor: using a cancer cell membrane coating that is identical (or close) to the tumor of interest and using antibodies or known ligands to target tumor specific antigens or surface molecules. Cancer cell membrane coatings are known to effectively encourage aggregation of nanoparticles at the desired tumor; however, their utility is very limited due to the cost and time required to recreate patient-specific cancer cell membranes [165]. Similarly, targeting tumor specific antigens is experimentally effective but requires extensive patient-specific preparation, which delays and prohibits utility. The more practical approach is perhaps to target the tumor endothelium using integrin-binding proteins, which are not unique to each patient and can therefore by definition be prepared in advance en masse.

Creating a nanoparticle surface chemistry that is too similar to the tumor will be immunogenic outside the immunoprotected tumor microenvironment. Therefore, measures to reduce immunogenicity must be undertaken to preserve the utility of the nanoparticle. PEGylation is perhaps the most utilized and well-studied technique for nanoparticle surface chemistry modification. PEGylation improves nanoparticle stability, increases hydrating capacity by creating a steric barrier, and delays identification and uptake by the reticuloendothelial system. It has been employed in numerous FDA-approved nanoparticle applications, from pancreatic cancer therapy to COVID-19 vaccines. The most studied pitfall of PEGylation is the generation of anti-PEG IgM antibodies, which reduce nanoparticle efficacy and can generate allergic reactions [166,167]. Another experimentally investigated approach is to use a red blood cell membrane coating, which has been demonstrated to encourage accumulation into tumor vasculature and to serve as a highly effective means of immune evasion [168].

Another way that nanoparticles can be modified to behave more specifically is to use stimuli-response systems. The most widely used stimuli-response system is by far pH and glutathione sensitive coatings for drug delivery, as these systems take advantage of tumor-specific metabolic conditions for activation of the nanoparticle [169,170,171]. Tumors thrive in relative hypoxia, and this has been an attractive target as well [172], mostly via nanoparticle “piggy-backing” on mesenchymal stem cells [173,174]. More recent experimental modalities include far red and near infrared sensitive nanoparticles that can be directed with light [175]. Additionally, 2% porphyrin-phospholipid photosensitizer liposomal coats increase drug delivery in laser-treated lesions [176]. Microwaves generate a hyperthermic environment that encourages nanoparticle tropism and structural changes that promote drug delivery; however, these systems have short circulation times [177]. Ultrasound responsive systems can be combined with MRI to disrupt tumor vasculature and increase drug extravasation in a very precise manner [178].

Several tumor pre-treatment strategies have been shown to improve nanoparticle accumulation. Most commonly, radiation therapy improves nanoparticle deposition and improves intratumoral distribution [179,180,181]. The use of anti-angiogenic drugs normalizes the vasculature in the microenvironment and decreases the pressure gradient in the interstitium, enhancing drug tissue penetration by reducing the gradient [182]. Pre-nanoparticle chemotherapy reduces the cell burden and increases tissue penetration by nanoparticles [183]. Laser ablation often causes vascular occlusion that prevents drug delivery, but the use of sub-ablative temperatures encourages perfusion and vascular permeability; combining sub-ablative laser therapy with radiation has been shown to be synergistic in enhancing nanoparticle response [184,185]. More experimental methods include pretreating the tumor with ECM-modifying enzymes to break down the physical barrier of the ECM and improve tissue penetration [186], as well as generating ROS local to the tumor using an external stimulating device, therefore reducing cell burden and increasing tissue penetration by nanoparticles [187].

## 5. Future Directions

Rapid and sensitive theranostic products are needed to reduce the tremendous patient cost and burden of brain cancers, as well as to reduce brain cancers’ impact on health care systems. A critical goal of future development will be the effective targeted delivery of imaging-active agents to the brain and specifically to the tumor cells. Both solid tumors as well as the margins after resection are in need of targeted imaging and delivery. As described, several nanoparticle systems have been developed, with promising results, and future areas still needing elucidation are exosomes and membrane vesicles where agents are packaged from the patient’s biochemical makeup.

The literature indicates the adaptability of conventional drug delivery systems used in brain delivery to accommodate the current theranostic agents. Nanoparticles are favored for brain tumor targeting due to the relatively simple preparation as well as functionalizability of the surface for targeting ligand. Table 1 gives examples of imaging-and treatment-focused nanoparticles from the literature. Polymeric systems using different polymer combinations including PLGA or phospholipids are combined to allow entrapment of the imaging or theranostic agent, such as gadolinium [161]. A gap in the approval process has been the safety concerns of the delivery systems using nanoparticles, with relatively few products approved considering when these systems were first identified. The polymer systems are well tolerated in vivo and have been shown to have favorable safety profiles. Of importance for this field, in fact, is that recently the increased safety profile of the delivery systems have been supported by the vaccine efforts of the SARS-CoV-2 COVID-19 epidemic since several of the RNA-based vaccines utilized PEG-based polymeric systems for delivery [188]. Targeting ligands are of interest, as they allow for cancer-specific uptake. For example, the increased expression of folate receptors could be targeted with folic acid-coated nanoparticles carrying a payload of gadolinium [162]. Peptide ligands have a strong history as targeting chaperones due to the specific interaction with surface receptors in cancer cells. For example, the group of Von Spreckelsen developed a targeting peptide to the brain-specific extracellular matrix protein brevican and used 18-F for PET imaging. As novel targets emerge from proteomica and other functional studies, current theranostic systems can be easily adapted to enhance treatment and imaging with quick turnaround time. Table 2 shows several of the FDA clinical trials that focus on theranostic development. As can be seen from the FDA site, there is a significant need for the development of new theranostic tools that target the CNS tumor scope, necessitating continued efforts to develop safe and effective theranostic agents and delivery methods. Some literature has provided evidence that there are possible solutions for improving delivery and decreasing toxicity of metal-based imaging agents, e.g., use of hydroxyapatite (HA) nanoparticles [189,190,191]. The HA is nontoxic and supports the effective pharmaceutical development of therapeutic agents. A better understanding of which targeting ligands, e.g., peptide or antibodies, will allow for selective payload delivery to tumor cells in the CNS is a key critical gap, alongside optimal chemotherapeutic or radio-ligand that will allow for minimal effective dosing of the area, achieving maximal cancer removal. As more of these products in the neuro-oncology space receive FDA approval, the barrier for entry into this therapeutic space will be simpler. Thus, it is important to improve our understanding of how to optimally localize and decrease CNS residence time for possible toxic compounds.

## Figures and Tables

**Figure 1 pharmaceutics-13-00948-f001:**
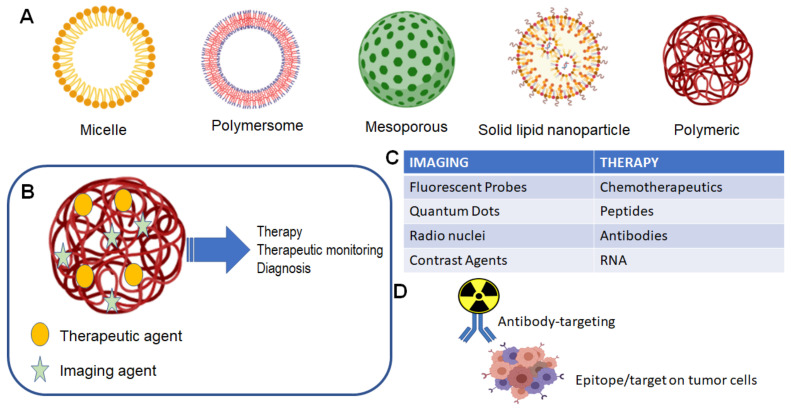
Nanoparticle and theranostic delivery. (**A**) Several nanoparticle systems have been described, each suited for optimal delivery and targeting of therapeutic agents [1,2]. (**B**) The theranostic nanoparticle can be formulated to include both an imaging agent and therapeutic agent [1]; (**C**) several options are available for either imaging of tumor and treatment of tumor [1]; (**D**) classical theranostic agent with radioactive cargo delivered to tumor via antibody targeting [3].

**Figure 2 pharmaceutics-13-00948-f002:**
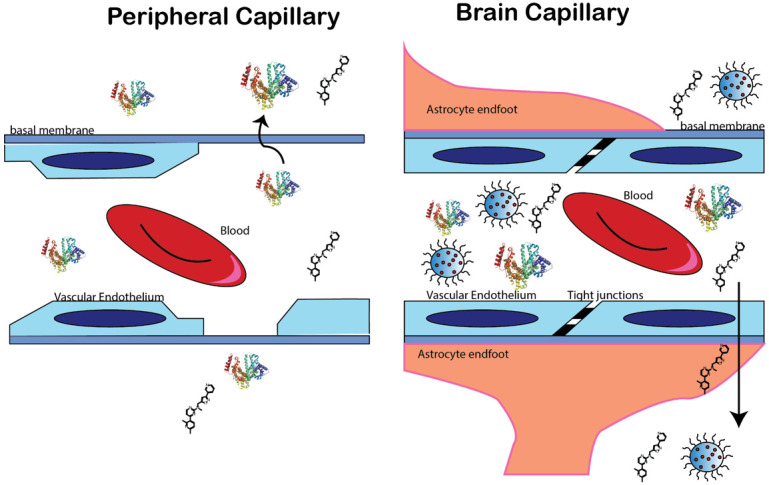
Delivery of theranostic molecules to the brain and tumor with use of nanoparticles. The blood–brain barrier (BBB) is a microvascular unit with tight junctions between the vascular endothelium cells causing selective permeability of small molecules. Use of nanoparticles can increase distribution of compounds to the brain with enhanced therapeutic or diagnostic outcomes.

**Table 1 pharmaceutics-13-00948-t001:** Therapeutic delivery of brain-targeted therapeutic/theranostic modalities [156,157,158,159,160,161,162].

Nanoparticle *	Cargo	Cancer
PLGA	Paclitaxel Methotrexate	Glioblastoma
PEG	DNA	Glioblastoma
PEG	Gadolinium/irradiation	Glioma
PEG/Phospholipids Targeting BRBP1 peptide	Iron oxide	Breast cancer brain metastasis
Somatostain peptide DOTA	Gadolinium	Brain cancers
Hyaloronic acid and angiopept-2	Gadolinium	Glioma
PEG and folate	Gadolinium	Cancers

* Primary polymeric system used in fabrication of nanoparticles.

**Table 2 pharmaceutics-13-00948-t002:** Theranostic-based clinical trials focused on brain or metastatic cancers.

Condition	Theranostic Agent	Title
Metastatic cancers (NCT04849247)	68Ga-DOTA-FAPI	68Ga-DOTA-FAPI and 177Lu-DOTA-FAPI Theranostic Pair in Patients With Various Types of Cancer (Locally Advanced or Metastatic Cancer)
Neuroendocrine tumors (NCT02609737)	68Ga-DOTA-JR11	Theranostics of Radiolabeled Somatostatin Antagonists 68Ga-DOTA-JR11 and 177Lu-DOTA-JR11 in Patients With Neuroendocrine Tumors
Neuroblastoma (NCT04023331)	67Cu-SARTATE	67Cu-SARTATE™ Peptide Receptor Radionuclide Therapy Administered to Pediatric Patients With High-Risk Neuroblastoma
Neuroblastoma (NCT01048086)	90Y-DOTA-tyr3-Octreotide	Theranostics: 68GaDOTATOC and 90YDOTATOC
Neuroendocrine (NCT02088645)	177Lu-PP-F11N	177Lu-PP-F11N for Receptor Targeted Therapy and Imaging of Metastatic Thyroid Cancer.
Glioblastoma Multiforme (NCT04373785)	Temozolomide; radiation	NG101m Adjuvant Therapy in Glioblastoma Patients

## Data Availability

Not applicable.

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
