# Peer review of "Nanoparticles as a Tool in Neuro-Oncology Theranostics"

_pharmaceutics, 2021, doi:10.3390/pharmaceutics13070948_

Round 1

Reviewer 1 Report

The authors have represented a review article entitled “Nanoparticles as a Tool in Neuro-Oncology Theranostics”. The manuscript prepared well and it will be an important review for the researchers who intend to work in this field. There are some points that need to be discussed before acceptance. 

  1. Please try to improve the Introduction section. In Introduction section authors need to explain in more detail similar types of reported work, with the novelty of the present work, limitations, and possibilities.
  2. Figure 1 need to be modified.
  3. Please make a table with FDA approved nanoparticles for neuro oncology theranostic.
  4. Please change the title “The Tumor Microenvironment” and discuss more briefly in this section. This section is deviating main goal of the review article. Authors need to ln the paragraph more carefully.
  5. Authors discussed the nanoparticles in their review manuscript are mostly toxic and needs to be modify with biocompatible coatings. Please include apatite-based hydroxyapatite-based nanoparticles for this type of treatment purpose. There are plenty of excellent research work already reported. Apatite based materials are biocompatible and have the efficiency for large drug loading and releasing.

Author Response

Please try to improve the Introduction section. In Introduction section authors need to explain in more detail similar types of reported work, with the novelty of the present work, limitations, and possibilities.

Sentence added to the end of the introduction

Figure 1 need to be modified.

Figure 1 has been modified as requested

Please make a table with FDA approved nanoparticles for neuro oncology theranostic.

There are no nanoparticles approved for theranostics in neuro-oncology

Please change the title “The Tumor Microenvironment” and discuss more briefly in this section. This section is deviating main goal of the review article. Authors need to ln the paragraph more carefully.

The section has been renamed

Authors discussed the nanoparticles in their review manuscript are mostly toxic and needs to be modify with biocompatible coatings. Please include apatite-based hydroxyapatite-based nanoparticles for this type of treatment purpose. There are plenty of excellent research work already reported. Apatite based materials are biocompatible and have the efficiency for large drug loading and releasing.

References to hydroxyapatite have been included

Reviewer 2 Report

The manuscript entitled “Nanoparticles as a Tool in Neuro-Oncology Theranosticsdescribed well research gap, literature review and data analysis study. This work is methodically carried out and scientifically correct. There are few issues that the authors can address to improve their manuscript before acceptance for publication.

What is the novel in this work? No previous literature published on nanoparticles as a tool for theranostic in neuro and oncology.

There is no mention of nanoparticles in figure 1. Please include in box.

The authors in section of future studies should talk about the gap in the knowledge and development in the field.

The authors should add their opinions and suggestions to stimulate more studies in the field.

The review is short on critique. A good review paper should discuss the barriers toward achieving the research goals in the field in conjunction with a summary of published reports.

The authors should include a reference from 2021 if possible.

Author Response

What is the novel in this work? No previous literature published on nanoparticles as a tool for theranostic in neuro and oncology.

This is a review article and therefore serves to review and discuss the role of nanoparticles as a tool in neuro-oncology theranostics.

There is no mention of nanoparticles in figure 1. Please include in box.

Figure 1 has been modified as requested

The authors in section of future studies should talk about the gap in the knowledge and development in the field.

This has already been included in the future directions section.

The authors should add their opinions and suggestions to stimulate more studies in the field.

This has been added to the future directions section.

The review is short on critique. A good review paper should discuss the barriers toward achieving the research goals in the field in conjunction with a summary of published reports.

This is adequately addressed in the section on toxicity, which highlights the barriers to utilization based on systemic toxicity of nanoparticles and the need for specific targeting.  Additional opinion has been added to the future directions section.

The authors should include a reference from 2021 if possible.

References from 2021 have been added throughout.